# Nanostructured Hybrid Hydrogels for Solar-Driven Clean Water Harvesting from the Atmosphere

**DOI:** 10.3390/ma15217538

**Published:** 2022-10-27

**Authors:** Md. Nizam Uddin, Md. Fozle Rab, A. K. M. Nazrul Islam, Eylem Asmatulu, Muhammad M. Rahman, Ramazan Asmatulu

**Affiliations:** 1Department of Engineering and Physics, Texas A&M University, Texarkana, TX 75503, USA; 2Bangladesh Railway, Rajshahi 6100, Bangladesh; 3Department of Mechanical System Engineering, Tokyo University of Agriculture and Technology, Tokyo 184-8588, Japan; 4Department of Mechanical Engineering, Wichita State University, Wichita, KS 67260, USA

**Keywords:** atmospheric water generator, hydrogel, relative humidity, electrospinning, fog harvesting

## Abstract

The scarcity of useable water is severe and increasing in several regions of the Middle East, Central and Southern Asia, and Northern Africa. However, the earth’s atmosphere contains 37.5 million billion gallons of water in the invisible vapor phase with fast replenishment. The United Nations Convention to Combat Desertification reports that by 2025 about 2.4 billion people will suffer from a lack of access to safe drinking water. Extensive research has been conducted during the last two decades to develop nature-inspired nanotechnology-based atmospheric water-harvesting technology (atmospheric water generator, AWG) to provide clean water to humanity. However, the performance of this technology is humidity sensitive, particularly when the relative humidity (RH) is high (>~80% RH). Moreover, the fundamental design principle of the materials system for harvesting atmospheric water is mostly unknown. In this work, we present a promising technology for solar energy-driven clean water production in arid and semi-arid regions and remote communities. A polymeric electrospun hybrid hydrogel consisting of deliquescent salt (CaCl_2_) and nanomaterials was fabricated, and the atmospheric water vapor harvesting capacity was measured. The harvested water was easily released from the hydrogel under regular sunlight via the photothermal effect. The experimental tests of this hybrid hydrogel (PAN/AM/graphene/CaCl_2_) demonstrated the feasibility of around 1.04 L of freshwater production per kilogram of the hydrogel (RH 60%). The synergistic effect enabled by photothermal materials and deliquescent salt in the hydrogel network architecture presents controllable interaction with water molecules, simultaneously realizing efficient water harvesting. This technology requires no additional input of energy. When considering the global environmental challenges and exploring the available technologies, a sustainable clean water supply for households, industry, and agriculture can be achieved from the air using this economical and practical technology.

## 1. Introduction

There is clear evidence that the water scarcity issue has been an increasing threat to the sustainable development of human society. The United Nations highlights the critical need of impoverished and developing regions (Middle East, Central and Southern Asia, and Northern Africa) of the world to achieve self-sustenance in the potable water supply in their millennium development goals [1]. World population growth, urbanization, and depleting water resources are the main driving forces for the rising global demand for water, and global climate change has also intensified this crisis, especially in countries with arid and semi-arid regions [2,3,4]. The concern is drastically increasing; therefore, scientists and engineers are challenged with urgently developing viable solutions to this problem soon. However, atmospheric water is a renewable water resource that is equivalent to ~10% of all freshwaters in lakes on earth [5]. About 4–25 g of water vapor is present in one cubic meter of air throughout the earth’s 100–600 m thick atmospheric boundary layer, which enables water to be supplied to any place on this globe [1]. Atmospheric water can be harvested in several ways: fog harvesting, active refrigeration, and sorption in conjunction with easily accessible low-grade energy. However, fog harvesting is an ancient method that requires constantly high ambient relative humidity (RH) [6,7]. Moreover, when the RH is low, many of the fog harvesting technologies are unusable. Therefore, in those areas that typically have low RH year-round and where harvesting atmospheric water is a necessity, harvesting water vapor from the air with a low RH is more meaningful than fog harvesting. However, until now, there has been very limited success achieved in harvesting water vapor from low RH air and subsequently delivering water with an easily accessible and self-sustained energy source [8,9]. To flourish the AWG technology, these three barriers (effectiveness at low RH, water quality, and low-temperature desorption) are needed to be overcome as shown in Figure 1.

Among the various methods of atmospheric water vapor harvesting, the sorption-based method is a new technology with the potential of being made appropriate, community-managed, and community-maintained in the context of developing countries. In this process, a solid sorbent is used to adsorb the water vapor from atmospheric air with the sorption heat being released and recovery of the extracted water by heating the adsorbent and condensing the water desorbed. This process can work at lower temperatures, and solar energy, which can be used to activate this process, is especially suitable for deserts or isolated islands that lack electricity [10]. The critical step for freshwater production based on the sorption process is the selection of solid sorbents. A good sorbent must possess a high water-uptake cycle, i.e., water sorption capacity, and the ability to store the adsorbed water until it is heated to desorb at low temperature as well. One fascinating advantage of the sorption-based atmospheric water vapor harvesting method is its capability of sorbing a large amount of water from dry air with humidity even lower than 20% and then releasing water at a relatively low temperature (70–90 °C) [11]. This low temperature can be achieved by using carbon-based photothermal materials such as carbon black, carbon nanotubes (CNTs), and graphene under regular or even weakened sunlight intensity [12,13,14,15,16,17]. Nevertheless, physical adsorption-based desiccants such as silica gel, clay, molecular sieves, and zeolite have wide water vapor sorption ability, but these necessitate higher temperatures (>160 °C) to effectively release the captured water [18,19].

Aristov et al. used their developed selective water sorbents and performed lab-scale tests for solar-driven freshwater production [20]. The selective water sorbents are a two-phase system that consists of a porous host matrix and an inorganic salt that is impregnated in pores. An external energy source is required for water desorption from selective water sorbents. Their test results demonstrated that about 3–5 tons of water per 10 tons of dry sorbent per day can be produced, but the equilibrium sorption time is as long as 60 h, and the desorption time is 30 h. Gordeeva et al. and Aristov et al. fabricated composite adsorbents by impregnating water-sorbing salts, i.e., calcium dichloride (CaCl_2_) or lithium bromide (LiBr), into porous media (SWS-1L, MCM-41) and investigated the water vapor adsorbing performance [21,22,23]. However, this method failed to achieve a high water-uptake cycle, mainly due to the small desorption at low temperatures. Later, a new adsorbent MCM-41/CaCl_2_ was used to manufacture a solar-driven water production unit [8]. The composite adsorbent was fabricated using an ultra-large pore crystalline material MCM-41 as the host matrix and CaCl_2_ as the hygroscopic salt. The adsorption capacity of these composites is 1.75 kg/kg dry adsorbent, which is higher than composites synthesized by silica gel and CaCl_2_. The developed unit exhibits the feasibility of freshwater production with daily water productivity of more than 1.2 kg/m^2^ of a solar collector area. Meanwhile, Kabeel theoretically and experimentally studied the performance of a system that consists of a sandy bed impregnated with a 30% concentration of CaCl_2_ to produce water from moist air [24]. A theoretical model was constructed to study the effect of various parameters such as solution concentration, temperature, and solar radiation intensity on the amount of collected water. This system could produce about 1.2 L of fresh water per square meter of glass cover per day. However, experimental results show an increase in the system productivity, i.e., 2.5 L of fresh water per square meter [25]. Wang et al. developed a novel composite adsorbent to increase the mass transfer area and the adsorption performance of the air-to-water system under hot and humid conditions [26]. Here, activated carbon fiber felt (ACF) was used as a host matrix. Experimental results showed that the ACF is more suitable as the matrix of composite adsorbents, and ACF30 has the best sorption performance of water uptake with 1.7 g/g capacity, which is three times more than silica gel-CaCl_2_. One drawback of this system is the deformation of the materials after sorbing water, which affects the structure of the absorbent bed. Later, the consolidation of ACF with lithium chloride (LiCl) was developed, with the advantages of a high water-uptake cycle and better structural properties [27].

In 2017, Kim and his co-workers demonstrated a porous metal-organic framework (MOF)-based device that captures water from the atmosphere at ambient conditions with relative humidity as low as 20% and delivers water using low-grade heat from natural sunlight (1 kW/m^2^) assisted by photothermal materials [11]. This device has a capacity of harvesting 0.25 L of water per kilogram of MOF daily at RH levels as low as 20% and requires no additional input of energy. The water vapor harvesting and releasing capacity of 14 common anhydrous and hydrated salts were investigated [28]. Among the salts screened, copper chloride (CuCl_2_), copper sulfate (CuSO_4_), and magnesium sulfate (MgSO_4_) distinguish themselves and are further made into bi-layer water-collection devices, with the top layer being photothermal and the bottom layer being a salt-loaded fibrous membrane. At a low relative humidity (down to 15%), these devices captured water vapor and released water under regular and even weakened sunlight (0.7 kW/m^2^). Gido et al. constructed a theoretical model for atmospheric water harvesting with a lithium chloride solution used as the water vapor sorbent. This system could produce water in a continuous process, and it reduced energy consumption by up to 65% compared with the conventional condensation-based system [29].

Additionally, deliquescent salt plays a major role in water sorption. It can sorb water vapor as many as 5–6 times its weight [30]. The water vapor is adequately adsorbed by inorganic salt because of the channel capillary force. The captured water vapor ultimately dissolves the salt and forms an aqueous solution with a much-expanded volume. In this study, an electrospun hybrid hydrogel composed of polyacrylonitrile (PAN), acrylamide monomer (AM), and multi-walled carbon nanotubes (MWCNTs)/graphene were fabricated with different proportions of PAN and AM and the atmospheric water vapor harvesting capacity was studied. The deliquescent salt was CaCl_2_. The easily expandable hydrogel network provides a perfect support platform that shows no constraint on water sorption performance. The novelty of the present work is to synthesize inexpensive solar-driven clean water harvesting methods from the atmosphere using multifunctional nanostructured materials and implement them for their use in real freshwater production systems.

## 2. Experiment

### 2.1. Materials

PAN (molecular weight, Mw 150,000 g/mole) was purchased from Sigma-Aldrich (St. Louis, MO, USA), and dimethylformamide (DMF) (99.8%) was purchased from Fisher Scientific. AM (99%), potassium persulfate (KPS) (99%,), N, N-methylenebis (acrylamide) (MBAA) (99%), and N, N, N, N-tetramethylethylenediamine (TEMED) (99%,) were purchased from TCI America. Sulfuric acid (H_2_SO_4_) (97%), nitric acid (HNO_3_) (70%), potassium hydroxide (1.0 N), MWCNTs with an outer diameter of 60–100 nm, and CaCl_2_ (99%) were purchased from Sigma-Aldrich and used without further purification. Graphene powder was purchased from Angstron Materials. The average X and Y dimensions of the graphene platelets were 14 µm, and the thickness was approximately 10–20 nm.

### 2.2. Functionalization of MWCNTs and Graphene

The MWCNTs were functionalized with a mixture of HNO_3_ and H_2_SO_4_ solutions. About 30 mL of HNO_3_ and 90 mL of H_2_SO_4_ were mixed, and 3.0 g of as-purchased MWCNTs were dispersed in the solution. The dispersion was then refluxed for 4 h at 70 °C followed by 2 h sonication. The as-treated dispersion was filtrated and washed thoroughly with deionized (DI) water before use. To functionalize the graphene, the required amount of graphene nanoflakes was placed in a conical flask. Then, sulfuric acid (97%), nitric acid (70%), and finally potassium hydroxide (1.0 N) solutions were added to the conical flux. The mixture was stirred at room temperature for a couple of hours. Then, the dispersion was strained using filter paper. The graphene nanoflakes were washed six to eight times with deionized water. Then, the dispersion was stirred overnight to ensure thorough dispersion of the graphene nanoflakes as well as to break down the larger graphene clusters. 

### 2.3. Fabrication of Electrospun PAN/AM//CaCl_2_ Hydrogel

The PAN and AM nanocomposite structures were fabricated using the electro-spinning technique under various voltages, collector distances, and pump speeds. The required amount (2 g of PAN and 2 g AM) of PAN and AM was dissolved in DMF (16 g by weight) and stirred at 40 °C for 4 h. The ratio of PAN and AM is 50:50. This ratio is optimized by several trials and the nanocomposites were successfully synthesized. If the quantity of AM was increased, then a thick viscous blend is formed, that is not spinnable. The solvent and polymer ratio was 80:20. The prepared solution was electro-spun and dried for 24 h in an open atmosphere. Then, the PAN/AM nanostructured materials were submerged in water (25 mL). The dispersion was purged with nitrogen for 1 h to eliminate any dissolved oxygen. Now, 0.4 g of KPS as initiator and 0.1 g of MBAA as a crosslinking agent were then added to the dispersion. Finally, 600 μL of TEMED solution was added as the cross-linking accelerator. The PAN/AM hydrogel was obtained after the mixture was settled overnight. The hydrogel was fabricated with/without MWCNTs and graphene incorporation. The as-prepared PAM/AM hydrogel was freeze-dried at −80 °C for 24 h. Then, 30 mL of CaCl_2_ solution with varying concentrations (0.2, 0.4, 0.6, 0.8 g/mL) was prepared and freeze-dried hydrogel was immersed in the CaCl_2_ solution for 24 h under ambient conditions. The as-prepared PAN/AM/CaCl_2_ hydrogel was dried at 80 °C in an oven for 3 days. The step-by-step PAN/AM/CaCl_2_ hydrogel fabrication process is presented in Figure 2. To incorporate the photothermal nanomaterials such as MWCNTs and graphene into the hydrogel, the PAN/AM nanostructured materials were submerged in a dispersion containing a specified amount of MWCNTs and graphene. In the present work, the MWCNTs and graphene loading in the hydrogel were set to 0.45 wt%. It was observed that a small amount of photothermal nanomaterials in the hydrogel would lead to a large increase in light absorbance [31].

### 2.4. Optimization of CaCl_2_ Loading

In this study, CaCl_2_ was loaded into the PAN/AM hydrogel using an impregnation of the aqueous CaCl_2_ solution. Digital photos of the PAN/AM hydrogels immersed in CaCl_2_ solutions at different concentrations are illustrated in Figure 3. As can be seen, after immersion for 24 h in the CaCl_2_ aqueous solution, the volume of the hydrogels expanded differently at different concentrations. In those that were treated with low concentrations (i.e., 0.2 and 0.4 g/mL), the volume expanded more, indicating high sorption of CaCl_2_. Based on our calculations, almost 80% of the CaCl_2_ was sorbed and retained inside the hydrogels, at a concentration of 0.4 g/mL, which is considerably higher when compared to the other options. In the case of high concentrations, less solution was incorporated into the hydrogel network because of the reduced swelling ratio of the PAN/AM hydrogel within a highly concentrated CaCl_2_ solution. The coordination between carbonyl (C=O) oxygens of AM and Ca^2+^ enhances the physical cross-linking of AM chains, and the salting-out effect causes the shrinkage of the AM’s hydrodynamic volume and overlap of the AM chains [32,33]. These two effects reduced the swelling ratio of the PAN/AM hydrogels. The results agreed with published articles in the literature, in which the solubility of AM in water decreases as a concentration of salt increases. [33,34] It was observed that the CaCl_2_ concentration of 0.4 g/mL yields the highest CaCl_2_ loading among all samples prepared. Thus, this concentration was optimized and chosen for all other samples studied here.

### 2.5. Material Characterization

The fabricated hydrogels were characterized using scanning electron microscopy (SEM) (FEI Nova Nano SEM 450) and a goniometer (water contact angles). With SEM, several images were taken of areas to inspect the porosity of the fabricated hydrogel. Before SEM, the samples were vacuum-dried and sputter-coated with a thin film of gold/palladium (Au/Pd) using a Leica EM ACE200 vacuum coater, since the samples were non-conductive. 

### 2.6. Solar Photothermal-Assisted Atmospheric Water Vapor Collection Device

The atmospheric water vapor sorption test was conducted using a homemade water collection device. Figure 4 shows a schematic of this easy-to-assemble homemade device for the present study. A magnifying glass was used to intensify the solar light onto the hydrogels. 

This atmospheric water vapor collection device consists of a plastic box, copper foil, aluminum foil, a transparent cap, and a commercial magnifying glass. The magnifying glass was used to concentrate the incoming sunlight, the aluminum foil was used as an isolator, and the copper foil was used as the heat-conductive sidewall. The copper foil has high thermal conductivity and therefore is beneficial to condense water on this wall plate surface where ambient air works as the heat sink. The temperature gradient between the cold copper surface of the container wall and the warm hydrogel regulates the water condensation process. When hydrogel is heated by the concentrated incoming sunlight, it produces water vapor with high partial pressure in the air inside the container, which is higher than the saturated vapor pressure of water. This water vapor meets the cold wall, condenses on its surface, and produces water. 

## 3. Results and Discussion

### 3.1. Morphology of PAN/AM/MWCNTs/CaCl_2_ Hydrogel

In the present work, a novel capacity-enhanced water vapor harvesting hydrogel from the atmosphere was presented. The sorbent was in the form of a hydrogel, with deliquescent salt (CaCl_2_) embedded inside the hydrogel. The deliquescent salt CaCl_2_ is responsible for water vapor capturing, and the cross-linked hydrogel network keeps the CaCl_2_ solution in a solid form, which enhances the water sorption capacity beyond conventional porous desiccants with rigid frameworks. Figure 5 presents SEM images of the hybrid hydrogel with graphene and MWCNTs inclusion. However, in the graphene hydrogel, graphene sheets are embedded into the polymer matrix. As can be seen, the hydrogel shows porous nanostructures with a low amount of micro and nanoscale fibers, which is critical for the absorption of atmospheric water. Measuring the porosity revealed that the average pore diameter is around 4 µm. The structural rigidity of the hydrogel can be controlled by the amount of the polymer in the hydrogel, which in turn can be controlled by changing the amount of PAN/AM monomer in the synthesis process. PAN and AM were chosen as the hydrogel network because of their mechanical and chemical stability, water-retaining ability, and low cost [35,36,37,38,39].

The hydrophobicity and hydrophilicity of a solid surface are mainly determined by the water contact angle (WCA) measurement. It is based on the observation of the intermolecular interactions between the surface and a small drop of water when the drop meets the surface. A contact angle larger than 90° indicates that the surface is hydrophobic. When the water contact angle is less than 90°, the surface is hydrophilic hence, the surface is wettable. Goniometer tests showed that the average water contact angle of the PAN/AM/CaCl_2_; PAN/AM/graphene/CaCl_2_ and PAN/AM/MWCNTs/CaCl_2_ hybrid hydrogel was 39 ± 2.1°, 41 ± 1.3° and 45 ± 1.8°, respectively. So, they are highly hydrophilic samples for better moisture absorption. Figure 6 shows the water contact angle of (a) PAN/AM/CaCl_2_; (b) PAN/AM/ graphene /CaCl_2_ and (c) PAN/AM/ MWCNTs/CaCl_2_ hybrid hydrogel.

### 3.2. Water Vapor Sorption and Release Assessment under Sunlight

The atmospheric water vapor using hydrogel was collected based on a homemade water vapor collection device. Several ways we can collect condensed water from the box. However, after condensation, we just open the box and collect the condensed water. Figure 7 shows digital photos of this device and the collected water.

The dry PAN/AM/CaCl_2_ hydrogel with a weight of 10 g was placed in the device, and the device was then placed outdoors overnight in Wichita, Kansas, USA, from 7 pm, 25 July 2020, to 7 am 26 July 2020. The average temperature and relative humidity values were 30 °C and 60%, respectively. The solar irradiation intensity of natural light was 6.21 kWh/m^2^/day and the hydrogel sorbed 11.6 g of water during a 12-h period. The water release and collection were conducted under natural sunlight without any concentration. This experiment was performed during 12:45–3:15 p.m., local time on 26 July 2020. Within this 2.5-h period, around 6.66 g of freshwater was collected inside the collection device. The reason that only 6.66 g of the 11.6 g of water was collected can be explained as follows: (a) a significant number of water droplets were sticking to the container walls, and (b) the water-releasing performance in the sealed chamber was lower because of the much higher water vapor partial pressure inside the sealed container. To further enhance the performance of the hydrogel, MWCNTs and graphene were incorporated into the hydrogel. Both were used as photothermal components because of their mechanical and chemical stability, superior light absorbance, outstanding light-to-heat conversion efficiency, and ability to localize thermal energy at the water-air interface, as reported in the literature [40,41]. In this study, the PAN/AM/MWCNTs/CaCl_2_ hydrogel and the PAN/AM/graphene/CaCl_2_ hydrogel were synthesized using in situ polymerization of electrosprayed PAN/AM nanostructures in the presence of MWCNTs and graphene. The main reason for adding MWCNTs and graphene to the hydrogel was to increase the light absorption capability. The water vapor sorption and water release experiments of the PAN/AM/MWCNTs/CaCl_2_ hydrogel and the PAN/AM/graphene/CaCl_2_ hydrogel were conducted on the same day as the experiment for the PAN/AM//CaCl_2_ hydrogel. Around 10 g of dried PAN/AM/MWCNTs/CaCl_2_ was placed in the collection device and then left outdoors overnight. The hydrogel sorbed 10.66 g of water during the 12-h period. The amount of sorbed water was slightly decreased due to the MWCNTs loading. The hydrogel is a porous structure and due to the geometry of nanoparticles, the addition of MWCNTs reduces the sorption kinetics. The water release and collection were conducted under natural sunlight. Within only a 2.5-h period, around 8.35 g of freshwater was collected inside the collection device. This can be attributed to the higher light absorbance and outstanding light-to-heat conversion efficiency of MWCNTs. The greater performance of the water release of the hydrogel can be attributed to the porous structure of the electrospun PAN/AM hydrogel, which is an effective strategy to increase the light absorption of carbon-based nanomaterials. However, the evaporate gas condensation ability and the droplet behavior greatly influence the water collection efficiency. Several studies investigated the effect of the thermal conductivity of the substrate materials on the fig collection efficiency [42]. Figure 8 illustrates the water uptake and release by the different types of hybrid hydrogel without using a magnifying glass. In a similar study, Li et al. fabricated a PAM/MWCNTs/CaCl_2_ hydrogel, and 35 g of this dry hydrogel was tested outdoors under field conditions, delivering 20 g of freshwater within 2.5 h under natural sunlight. [43] However, water uptake and release for the graphene-incorporated hydrogel were 10.79 g and 8.85 g, respectively, for 10 g of dried PAN/AM/graphene/CaCl_2_ hydrogel for the same duration of water uptake and release. A magnifying glass was used in the homemade collection device to enhance further water release from the hydrogel. Figure 9 shows the performance of the water release using a magnifying glass. With the magnifying glass, the water release rate was substantially increased for all hybrid hydrogels studied here.

As can be seen from Figure 9, about 97% of the sorbed water is released from the hybrid hydrogel containing graphene. Using a magnifying glass in the collection device produces more water than without it. The magnifying glass concentrated the sunlight and produced a higher temperature of the hydrogel, which enhanced the water release. With a magnifying glass, the PAN/AM/graphene/CaCl_2_ hydrogel can produce around 1.04 L of freshwater per kilogram of hydrogel (RH 60%). Figure 10 presents a digital photo of the PAN/AM/graphene/CaCl_2_ hydrogel water collection and condensed water on the wall of the box during the water collection process.

The stability and reusability of the fabricated hybrid hydrogel were studied using multi-cycle water vapor harvesting tests. One cycle of water collection included 12 h of the water uptake and 2.5 h of the water release process using the previously mentioned method, followed by air-drying, and drying in an oven for 30 min at 60 °C. The water vapor collection performance of the hydrogel was recorded after 5, 10, and 15 cycles. Figure 11 shows the water collection of the PAN/AM/graphene/CaCl_2_ hydrogel for different water collection cycles (with magnifying glass). As can be seen, there was no obvious impairment in the water collection performance, even after 15 cycles of repetitive water collection. Like PAN/AM/graphene/CaCl_2_ hydrogel, the water vapor collection of PAN/AM/MWCNTS/CaCl_2_ hydrogel was also studied. Table 1 summarizes the water collection performance of PAN/AM/MWCNTS/CaCl_2_ hydrogel multi-cycle water vapor harvesting tests. Even after 15 cycles of repetitive water collection cycles, the hydrogel exhibits a stable water collection performance. This can be attributed to the structural rigidity of the hydrogel and the retainability of the CaCl_2_ salt in the hydrogel network. Even after 15 cycles, a stable water uptake and release performance was observed. 

The water vapor sorption characteristics of the PAN/AM/CaCl_2_ hydrogel can be explained by the phase diagram of water-CaCl_2_ [44]. The CaCl_2_ contains two primary stages in the water sorption process. First, through a hydration reaction, an anhydrous CaCl_2_ crystal captures water molecules and forms hydrates for better water collection. Second, when the CaCl_2_ sorbs enough water, it forms CaCl_2_·6H_2_O, which in turn is dissolved in the sorbed water as more water is sorbed. The vapor pressure of a saturated CaCl_2_ aqueous solution at 25 °C is 0.9 kPa, which is equivalent to 26% relative humidity. In other words, the water sorption by CaCl_2_ at an RH < 26% is attributed to its increase in the hydration water, and that which occurred at RH > 26% leads to a dilution of the CaCl_2_ aqueous solution, i.e., deliquescence. This RH of 26% is a critical point that can be varied with the ambient temperature. Both the PAN/AM/MWCNTs/CaCl_2_ and PAN/AM/graphene/CaCl_2_ hydrogels are suitable candidates for freshwater harvesting from the atmosphere in an arid region, or in the case of an emergency response or natural disaster, war, etc. Due to some factors such as daily variations in solar irradiation density and weather conditions, the performance of this water collection device may vary. However, some simple maintenance such as cleaning and wiping the collection box is necessary from time to time in operating this device. 

The World Health Organization (WHO) has suggested that the minimum water intake for an individual is 3 L per day per person, and this type of device is good for providing such freshwater for a household with two members (a total of 6 L) per day. The salt used in this study is CaCl_2_, which is eco-friendly and non-toxic. This sunlight-assisted atmospheric water collection device is focused on providing clean water to fulfill the minimum water intake requirement in a household with two members.

## 4. Conclusions

In the present study, a green and low-cost and practical technology for an atmospheric water harvesting system was developed for the collection of atmospheric freshwater. The PAN/AM/CaCl_2_ hydrogel was fabricated with and without MWCNTs and graphene, and water vapor harvesting performance was investigated. An “easy-to-assemble” water collection device was fabricated to measure the water collection performance of the fabricated hydrogel. The experimental results of this water collection indicate that without MWCNTs/graphene, the hydrogel can sorb 1.16 g of water per gram of hydrogel, and under natural sunlight, it can release 0.66 g of freshwater per gram of water-loaded hydrogel. However, the incorporation of MWCNTs slightly reduced the water-uptake capacity but increased the water release rate. Around 1.06 g of water per gram of hydrogel was sorbed, and water release was increased to 0.78 g per gram of water-loaded hydrogel. Moreover, the incorporation of graphene in hydrogel enhances the water released from the hydrogel. The hydrogel containing graphene sorbed 1.07 g of water per gram of hydrogel and released 0.82 g of freshwater per gram of water-loaded hydrogel. This can be attributed to the greater light absorbance and outstanding light-to-heat conversion efficiency of the graphene and MWCNTs. Additionally, the integration of a magnifying glass in the collection device greatly ameliorated the water harvesting performance. The magnifying glass concentrated the sunlight and increased the hydrogel temperature in the collection device, and about 97% of the sorbed water was released from the hybrid hydrogel containing graphene. Moreover, the fabricated hybrid hydrogel can be used repeatedly for atmospheric water vapor collection without a decline in its performance and is suitable for a wide range of humidity levels. These hydrogels are inexpensive, require no additional input of energy, and are especially suitable for clean water production from the atmosphere, such as in arid and semi-arid areas for drinking, agriculture, and industrial purposes.

## Figures and Tables

**Figure 1 materials-15-07538-f001:**
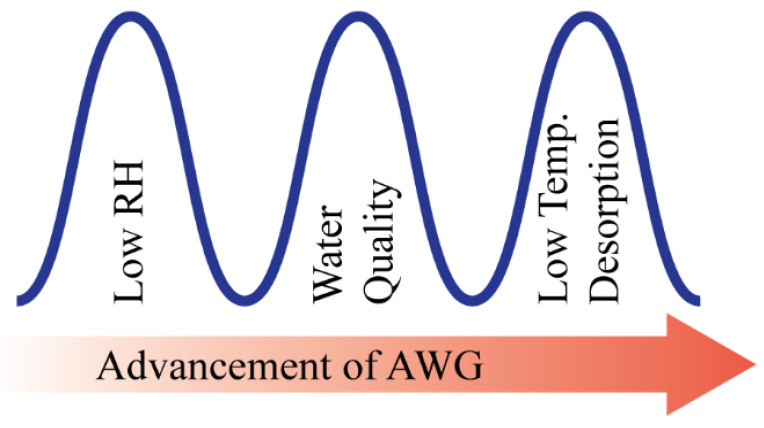
Barriers to the progress of AWG technology.

**Figure 2 materials-15-07538-f002:**
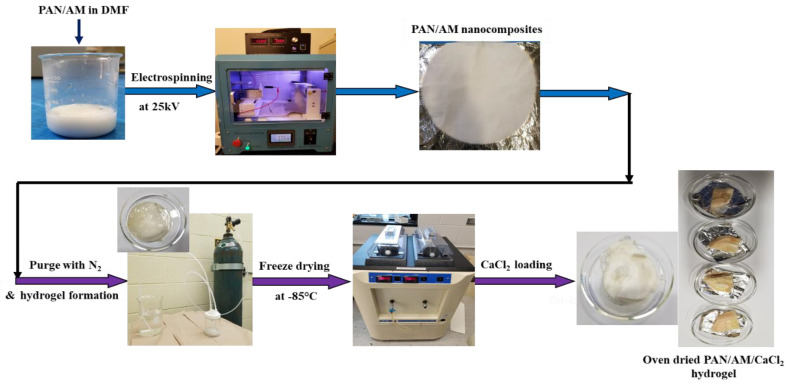
Step−by−step fabrication of PAN/AM/CaCl_2_ hydrogel for atmospheric water collection.

**Figure 3 materials-15-07538-f003:**
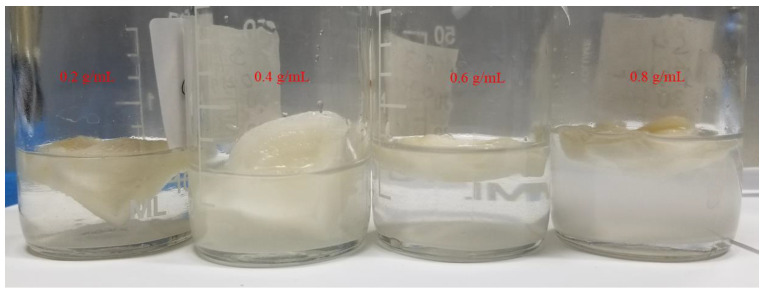
Digital photos of PAN/AM hydrogels immersed in different concentrations of CaCl_2_ solutions.

**Figure 4 materials-15-07538-f004:**
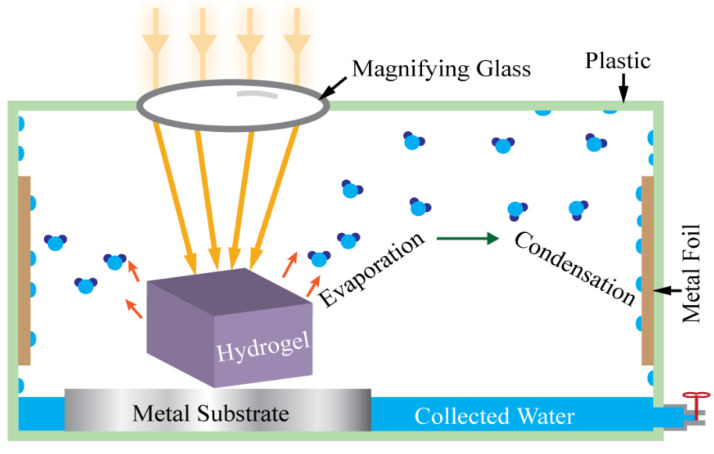
Schematic view of solar photothermal-assisted atmospheric water vapor collection device.

**Figure 5 materials-15-07538-f005:**
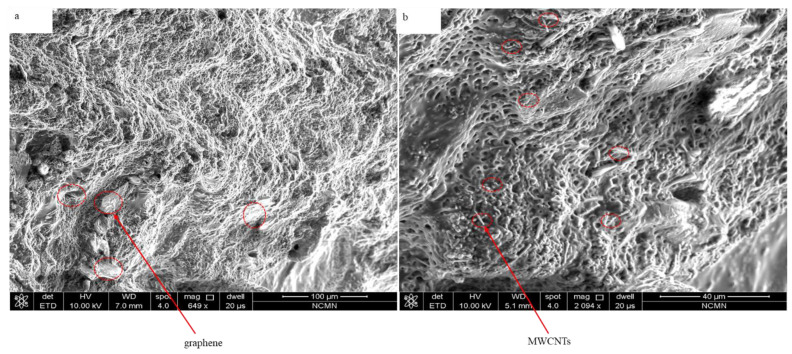
SEM images of hybrid hydrogel: (**a**) PAN/AM/graphene/CaCl_2_, and (**b**) PAN/AM/MWCNTs/CaCl_2_.

**Figure 6 materials-15-07538-f006:**
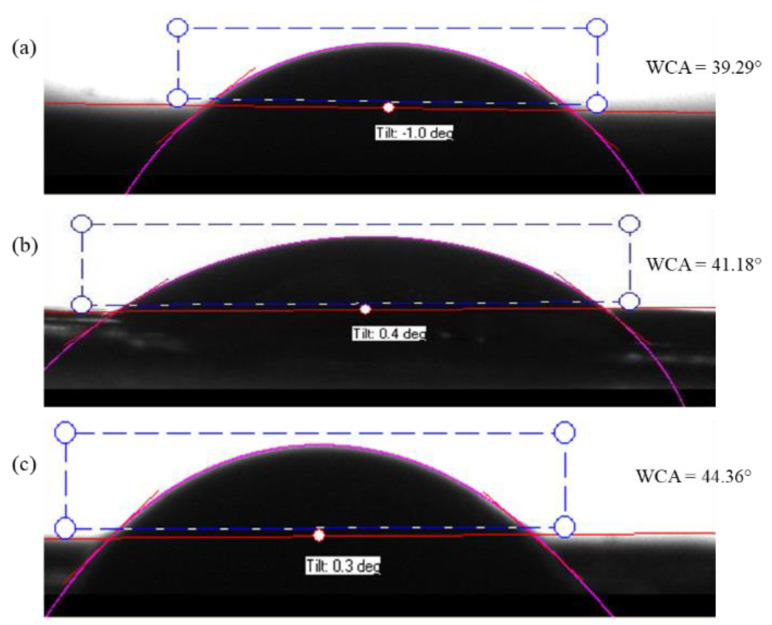
Water contact angle of the hydrogel; (**a**) PAN/AM/CaCl_2_ (**b**) PAN/AM/graphene/CaCl_2_ and (**c**) PAN/AM/MWCNTs/CaCl_2_.

**Figure 7 materials-15-07538-f007:**
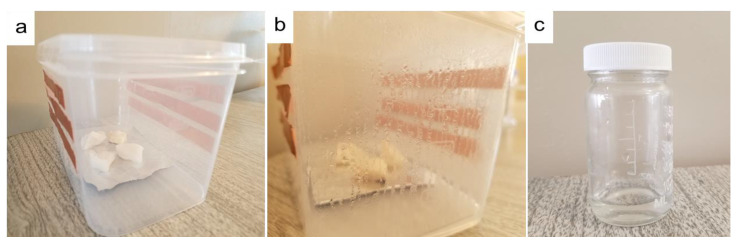
Digital photos of water collection: (**a**) homemade water collection device prototype based on PAN/AM/CaCl_2_ hydrogel; (**b**) condensed water on the wall of the box during the water collection process; and (**c**) water collected from a plastic box.

**Figure 8 materials-15-07538-f008:**
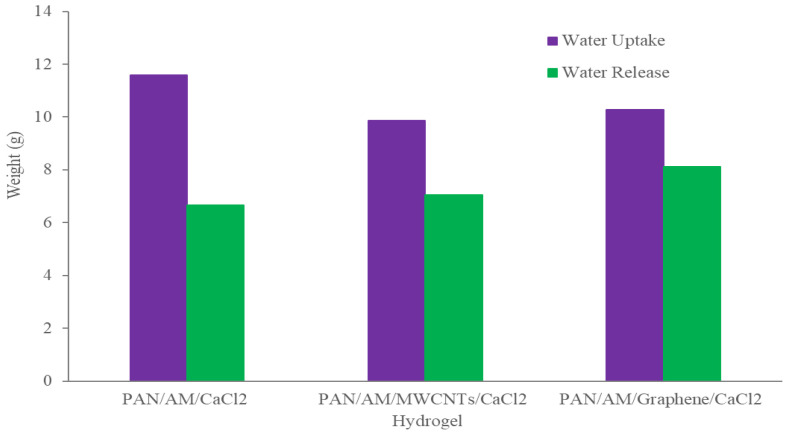
Atmospheric water collection using hydrogel without a magnifying glass.

**Figure 9 materials-15-07538-f009:**
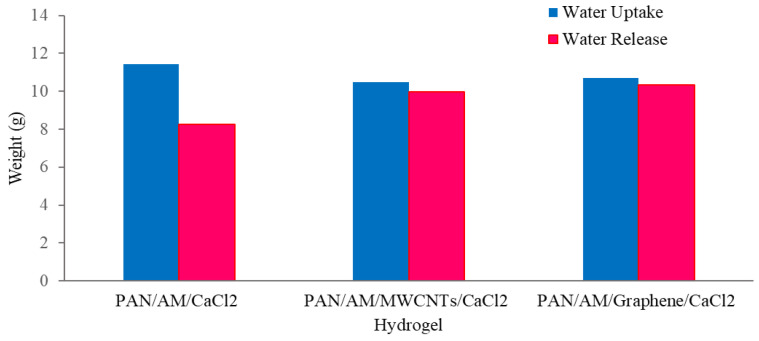
Atmospheric water collection using hydrogel with a magnifying glass.

**Figure 10 materials-15-07538-f010:**
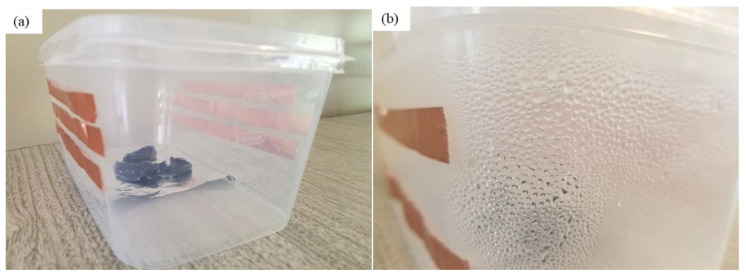
Digital photos of water collection: (**a**) device prototype based on PAN/AM/graphene/CaCl_2_ hydrogel; and (**b**) condensed water on the wall of the box during the collection process.

**Figure 11 materials-15-07538-f011:**
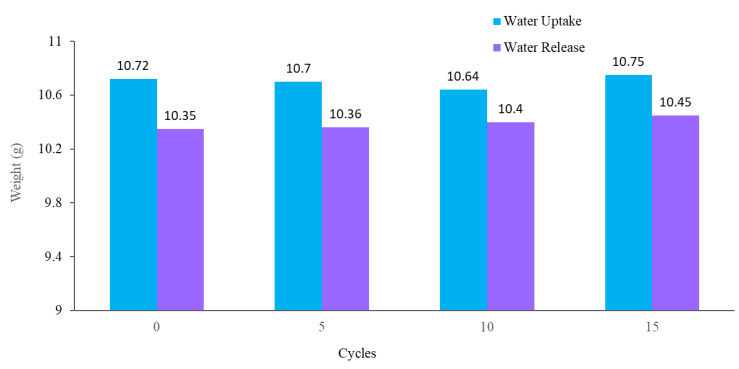
Water collection of PAN/AM/graphene/CaCl_2_ for different water collection cycles (with magnifying glass).

**Table 1 materials-15-07538-t001:** Water collection of PAN/AM/MWCNTs/CaCl_2_ for different water collection cycles (with magnifying glass).

Cycle	Water Uptake (g)	Water Release (g)
0	10.48	9.96
5	10.42	9.90
10	10.53	10.03
15	10.5	9.99

## Data Availability

Not Applicable.

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
