# Peer review of "Nanostructured Hybrid Hydrogels for Solar-Driven Clean Water Harvesting from the Atmosphere"

_materials, 2022, doi:10.3390/ma15217538_

Round 1
Reviewer 1 Report
The manuscript presents an electrospun hydrogel loaded with CaCl2 and CNT or graphene for atmospheric water harvesting (AWH) applications. They reported good water production (1.04 liters of water per kilogram of the hydrogel at RH 60%) and claimed it to be a promising and inexpensive technology. I have the following comments and questions for the authors.
1. The novelty of the paper is not clear. AWH has been widely studied in recent years. The concepts of using hydrogels, electrospun fibers, CaCl2 as the sorbent, solar-driven methods, and CNT/graphene for enhanced solar heating are all not new (https://doi.org/10.1016/j.joule.2021.04.005). Some recent system has reported very similar PAM-CNT-CaCl2 gel design (https://doi.org/10.1038/s41893-020-0535-4). Their materials characterizations also did not show new scientific insights into the materials system or AWH mechanism.
2. The authors claimed that the technology is inexpensive and estimated the cost of the materials. However, the electrospinning process is high-cost, compared to other film production processes. Why do we use electrospinning and nanofibers? Although the authors claim it enhances the water sorption capacity, the result is not better than similar films without nanostructures (https://doi.org/10.1038/s41893-020-0535-4). The authors should at least compare their nanofibers with a film with the same composition produced without electrospinning in their lab. They should also remove the cost estimation part if the electrospinning cost is not considered.
3. The kinetics of water sorption and desorption are critical for AWH applications. The authors should provide the water sorption/desorption rates at various humidity and temperature to compare with the literature.
Author Response
Dear Professor,
I uploaded a revised version of the manuscript with addressing the review comments. Please see our answers to the review comments and let us know if we can do any additional corrections and clarifications.
Sincerely,
Md. Nizam Uddin, Ph.D.
Assistant Professor of Mechanical Engineering
Department of Engineering and Physics
Texas A & M University-Texarkana
7101 University Ave
Texarkana, TX 75503
Email: muddin@tamut.edu
Cell: (870) 621-4437

Reviewer 2 Report
Uddin et al. report on the synthesis of new hydrogels based on an interpenetrated polymer network and their use for water-harvesting. The results are relatively promising in terms of water-harvesting and photothermal desorption and the paper will be of interest to the readership of Materials. However, further details and characterizations of the materials is still required:
- Experimental part: precisions are required (volume of DMF, water, concentration of the graphene or MWCNTs solution...
- How is the MWCNTs/graphene loading controlled?
- The polymer matrix should be characterized (for instance using FTIR) to confirm the chemical structure and incorporation of the polyacrylamide.
- thermogravimmetric analysis (TGA) measurements should be performed to confirm CaCl2 and MWCNTs or graphene content.
- SEM images (Fig. 5): a SEM image of the PAN/AM with graphene should be provided. Also, the images should be at the same scale for comparison.
- Contact angle measurements (Fig. 6): the control experiment without graphene/MWCNTs should be added.
Author Response

(The authors gave the same response as above.)

Author Response

(The authors gave the same response as above.)

Round 2
Reviewer 1 Report
The authors have not addressed my previous comments, and therefore I cannot agree to accept the manuscript.
1. My previous comment 1 asked about the novelty of their work. They did not state it clearly in their manuscript. Their response did not convince me, and they did not revise the manuscript accordingly. Their response claims that they focus on "simple and more affordable materials and approaches", which is not true compared with the references I mentioned. Also, I did not find evidence for the claim that their approach is "more consistent and technologically more appropriate for the future practical use".
2. The authors did not fully address my previous comment 2. The authors should at least compare their nanofibers with a film with the same composition produced without electrospinning in their lab.
3. The authors did not provide any new information about the kinetics of water sorption and desorption as requested in my previous comment 3, which are basic tests in this area.
Author Response
Dear Reviewers,
We are very grateful for the valuable recommendations, which further increased the value of our manuscript. Please see our answers to your comments and let us know if we can do any additional corrections and clarifications.
Sincerely,
Md. Nizam Uddin, Ph.D.

Reviewer 2 Report
The paper can be published in present form.
Author Response
N/A